# Role of B Cell-Activating Factor in Fibrosis Progression in a Murine Model of Non-Alcoholic Steatohepatitis

**DOI:** 10.3390/ijms24032509

**Published:** 2023-01-28

**Authors:** Kozue Kanemitsu-Okada, Masanori Abe, Yoshiko Nakamura, Teruki Miyake, Takao Watanabe, Osamu Yoshida, Yohei Koizumi, Masashi Hirooka, Yoshio Tokumoto, Bunzo Matsuura, Mitsuhito Koizumi, Yoichi Hiasa

**Affiliations:** Department of Gastroenterology and Metabology, Ehime University Graduate School of Medicine, To-on 791-0295, Japan

**Keywords:** B cell-activating factor, non-alcoholic fatty liver disease, non-alcoholic steatohepatitis, liver fibrosis, hepatic inflammation, macrophage, stellate cell

## Abstract

Non-alcoholic fatty liver disease (NAFLD) is the most prevalent chronic liver disease all over the world. Therapeutic strategies targeting its multidirectional pathways are required. Particularly, fibrosis is closely associated with its prognosis. We previously found that B cell-activating factor (BAFF) is associated with severity of NAFLD. Here, we determined the direct in vivo role of BAFF in the development of liver fibrosis. Histological and biochemical analyses were performed using wild-type and BAFF-deficient mice. We established a murine model of non-alcoholic steatohepatitis (NASH) using carbon tetrachloride injection accompanied by high-fat/high-cholesterol diet feeding. Additionally, in vitro analysis using mouse macrophage-like cell line RAW264.7 and primary hepatic stellate cells was performed. Hepatic steatosis and inflammation, and most importantly, the progression of liver fibrosis, were ameliorated in BAFF-deficient mice compared to those wild-type mice in our model. Additionally, BAFF deficiency reduced the number of CD11c^+^ M1-type macrophages in the liver. Moreover, BAFF stimulated RAW264.7 cells to secrete nitric oxide and tumor necrosis factor α, which drove the activation of hepatic stellate cells. This indicates that BAFF plays a crucial role in NASH development and may be a promising therapeutic target for NASH.

## 1. Introduction

Non-alcoholic fatty liver disease (NAFLD) is an exhibit of systemic disease based on lifestyle-related diseases such as obesity, diabetes, and metabolic syndrome. The number of patients with NAFLD is rapidly increasing with the rising prevalence of obesity, and it has become a global health problem [1,2]. NAFLD is not just a manifestation of a metabolic syndrome but has emerged as a driver of systemic disease [3]. Previous studies have suggested that the fibrosis stage is an independent and significant predictor of overall mortality and liver-related events [4,5,6]. Therefore, it is critical to identify the drivers that mediate the progression of non-alcoholic fatty liver (NAFL) to non-alcoholic steatohepatitis (NASH).

As chronic hepatic inflammation has been shown to contribute to disease progression, one of the potential therapeutic targets of NAFLD is the suppression of inflammation and fibrosis by the regulation of immune abnormalities. We have been conducting research on the immunological aspects of NAFLD [7,8,9] development. B cell-activating factor (BAFF) (CD257) is a factor that promotes the expansion and differentiation of the B cell population [10,11]. BAFF belongs to the tumor necrosis factor (TNF)-ligand family and is secreted from activated T cells, B cells, and myeloid lineage cells such as macrophages and dendritic cells. Previously, we had reported that serum BAFF levels were increased, and that BAFF was preferentially expressed, in the visceral adipose tissue of high-fat diet (HFD)-induced obese mice [12,13,14]. In addition, hepatic steatosis was attenuated in BAFF-deficient mice fed an HFD compared to in control mice [15]. Furthermore, serum BAFF levels in patients with NASH were higher than those in patients with NAFL [16]. Collectively, these data indicate that BAFF may be closely related to hepatic steatosis, inflammation, and fibrosis in NASH. However, the direct role of BAFF in hepatic fibrosis has not yet been elucidated.

In this study, we established a murine model of NASH that led to the reproducible progression of steatohepatitis with significant fibrosis using carbon tetrachloride (CCl_4_) injection accompanied by high-fat/high-cholesterol diet (HFHCD) feeding. To determine the in vivo role played by BAFF in the development of liver fibrosis, we studied the NASH model using BAFF-deficient mice and found that BAFF-deficient mice were protected from developing not only steatosis but also NASH and fibrosis. Our findings indicate that BAFF plays an important role in the development of liver fibrosis and may be a therapeutic target for NASH.

## 2. Results

### 2.1. BAFF Deficiency Attenuates Liver Steatosis in Murine Models of NASH

In previous studies [13,14,15], we clarified the role of BAFF in a mouse model of NAFLD that was fed with HFD; however, there was mild inflammation and little fibrosis in these mice. To explore the role of BAFF in liver fibrosis in NASH, we established a new mouse model using HFHCD/CCl_4_-induced liver injury (Figure 1a), which was modified from previous reports [17,18]. The serum BAFF levels (11,080.7 ± 187.8 pg/mL: *n* = 10) in HFHCD/CCl_4_-treated wild-type mice were significantly higher than those fed normal diet chow (7718.3 ± 318.1 pg/mL: *n* = 10: *p* < 0.01).

As previously shown in mice with normal diet (ND) chow [15], there was no difference in body weight gain between the HFHCD/CCl_4_-treated wild-type (WT) and B6.129S2-Tnfsf13b^tm1Msc^/J (*BAFF^−/−^*) mice. However, liver weight was significantly higher in HFHCD/CCl_4_-treated *BAFF^−/−^* mice than in HFHCD/CCl_4_-treated WT mice (Figure 1b). Serum levels of aspartate aminotransferase (AST) and alanine aminotransferase (ALT) were significantly lower in HFHCD/CCl_4_-treated *BAFF^−/−^* mice than that in HFHCD/CCl_4_-treated WT mice (Figure 1c). Histological analysis revealed that hepatic fat accumulation was lower in HFHCD/CCl_4_-treated *BAFF^−/−^* mice than in HFHCD/CCl_4_-treated WT mice (Figure 1d). Furthermore, hepatic triglyceride and total cholesterol levels were significantly lower in HFHCD/CCl_4_-treated *BAFF^−/−^* mice than in HFHCD/CCl_4_-treated WT mice (Figure 1f).

### 2.2. Liver Inflammation Is Attenuated in BAFF^−/−^ Mice in Murine Models of NASH

In addition to steatosis, liver inflammation was attenuated in HFHCD/CCl_4_-treated *BAFF^−/−^* mice compared to in HFHCD/CCl_4_-treated WT mice (Figure 1d,e). We performed immunohistochemical and flow cytometric analyses of immune cell populations and assessed inflammation-related gene expression in the liver by real-time RT-PCR. In immunohistochemical analysis, the number of macrophages and crown-like structures (CLSs) in the liver was significantly lower in *BAFF^−/−^* mice than that in WT mice treated with HFHCD/CCl_4_ (Figure 2a). Flow cytometric analysis showed that the proportion of F4/80^+^CD11c^+^ M1-like macrophages in the livers of HFHCD/CCl_4_-treated *BAFF^−/−^* mice was significantly lower than that in HFHCD/CCl_4_-treated WT mice (Figure 2b). In addition, the expression of *TNF-α*, *interleukin (IL) 6, monocyte chemotactic protein (MCP)-1,* and M1-macrophage-related markers, such as *CD11c* and *inducible nitric oxide synthase* (*iNOS*), was significantly lower in the livers of *BAFF^−/−^* mice than that in WT mice treated with HFHCD/CCl_4_ (Figure 2c). The expression of iNOS in macrophages was confirmed by immunohistochemistry, and iNOS-positive areas in the livers of HFHCD/CCl_4_-treated *BAFF^−/−^* mice were significantly lower than in HFHCD/CCl_4_-treated WT mice (Figure 2d). These data suggest that attenuated liver inflammation in BAFF deficiency is associated with a reduction in proinflammatory cytokines produced by macrophages in our models.

### 2.3. BAFF-Deficient Mice Are Protected from the Development of NASH and Fibrosis

Furthermore, we investigated the role of BAFF in liver fibrosis using our NASH model. Sirius Red (SR) staining revealed extensive interstitial fibrosis in the livers of HFHCD/CCl_4_-treated WT mice, which was substantially ameliorated in HFHCD/CCl_4_-treated *BAFF^−/−^* mice (Figure 3a). We confirmed that the total collagen content in the livers of HFHCD/CCl_4_-treated *BAFF^−/−^* mice was significantly lower than in HFHCD/CCl_4_-treated WT mice (Figure 3b). Additionally, the number of α-smooth muscle actin (SMA) antibody (+) cells was lower in HFHCD/CCl_4_-treated *BAFF^−/−^* mice than in HFHCD/CCl_4_-treated WT mice (Figure 3c). With respect to gene expression, the mRNA expression of *transforming growth factor (TGF)-β1* and *collagen (Col)-1a1* was significantly lower in the livers of *BAFF^−/−^* mice than in WT mice treated with HFHCD/CCl_4_ (Figure 3d).

To test whether the role of BAFF is limited to the specific case of HFHCD/CCl_4_-induced liver injury, we used two alternative model systems: (1) feeding with HFHCD for a longer period (Figure 4a) and (2) feeding with choline-deficient, L-amino-acid-defined high-fat diet (CDAHFD) (Figure 4b). In both models, collagen deposits in the liver were significantly lower in *BAFF^−/−^* mice than in WT mice (Figure 4c,d). Thus, liver fibrosis progression was ameliorated in the absence of BAFF in at least three model systems.

### 2.4. Hepatic Stellate Cell Activation Is Driven by Interaction with Macrophages in Murine Models of NASH

A crucial downstream consequence of hepatic inflammation is the activation of hepatic stellate cells (HSCs), the principal fibrogenic cell type in the liver. To explore the underlying mechanisms of the differences in liver fibrosis in our models, we investigated the direct effects of BAFF on HSCs. However, receptors for BAFF, such as BAFF-receptor (BAFF-R), ransmembrane activator, calcium-modulator, and B-cell maturation antigen, were not expressed in primary cultured mouse HSCs. Furthermore, the addition of BAFF to HSCs in culture did not alter the gene expression of HSCs such as *α-SMA*. These data indicate that BAFF has little or no direct effect on HSCs.

Thereafter, we focused on macrophages, which are one of the main sources of potent pro-fibrogenic signals [19]. In addition to hepatocytes [13], macrophages express BAFF-R [20].

Hence, we examined whether BAFF-R exerted its signaling function in macrophages and found that nuclear factor (NF)-κB2 and RelB, the non-canonical NF-κB pathway, were induced in the mouse macrophage-like cell line (RAW264.7) treated with BAFF (Figure 5a). Furthermore, as shown in Figure 5b, BAFF treatment upregulated the expression of genes encoding proteins related to inflammation and tissue repair, including *iNOS* and *IL-10*. These changes were inhibited by treatment with the recombinant BAFF-R Fc (Figure 5b).

Finally, we investigated whether the effect of BAFF on macrophages influenced HSC activation. Isolated mouse HSCs were cultured with conditioned medium (CM) from BAFF- or phosphate-buffered saline (PBS)-treated-RAW264.7 cells in addition to lipopolysaccharide (LPS). The mRNA levels of *TGF-β1* and *α-SMA* were significantly increased in HSCs following culture with CM from BAFF-treated-RAW264.7 cells (Figure 6a). Moreover, nitric oxide (NO) (LPS 344.7 ± 94.08 μM, LPS+BAFF 423.4 ± 83.28 μM: *p* < 0.05) and TNF-α levels in CM from BAFF-treated-RAW264.7 cells were significantly higher than those from PBS-treated RAW264.7 cells (Figure 6b). However, the IL-6 levels in CM were not different between the two groups. These results indicate that BAFF-treated macrophages generate inflammatory cytokines such as NO and TNF-α and activate HSCs, leading to the development of liver fibrosis in NASH.

## 3. Discussion

The prevalence of NAFLD is increasing and liver-related morbidity and mortality will dramatically increase worldwide within the next few decades [2]. At present, lifestyle modification is the mainstay of therapeutic recommendations and no specific pharmacological treatment is available for NAFLD [21]. Drugs targeting metabolism, inflammation, and fibrogenesis are under development; however, several compounds have only improved the investigational endpoints in a subset of patients who are exposed to these drugs, indicating that targeting a single pathway may be insufficient [22]. To overcome these problems, combination therapies and/or therapies targeting multidirectional effects may be promising approaches.

We previously found that BAFF is associated with NAFLD severity in Japanese patients [16]. Furthermore, BAFF deficiency prevents fat accumulation in the liver by suppressing the visceral adipose tissue inflammation and de novo lipogenesis in an HFD-fed mouse model [15]. Although liver fibrosis was not observed in liver specimens from HFD-fed mice, the expression of fibrosis-related genes, such as *TGF-β1*, *α-SMA*, and *Col-1a1*, was significantly lower in the livers of *BAFF^−/−^* mice than in those of WT mice.

Furthermore, BAFF promotes collagen production by dermal fibroblasts from patients with systemic sclerosis [23], and BAFF inhibition attenuates skin and liver fibrosis in mouse models of scleroderma [24]. These findings suggest that BAFF may play a role not only in metabolism and inflammation but also in fibrosis in the pathogenesis of NASH, which indicates that it is one of the therapeutic targets in NASH.

In the present study, we report a new murine NASH model using HFHCD and CCl_4_, which develops the histological features of NASH with extensive fibrosis and severe inflammation. HFHCD has been widely used to establish mouse NASH models; however, the major disadvantage of this NASH model is that it does not fully progress to severe steatohepatitis, even after long-term feeding. CCl_4_ has been traditionally used for decades to induce liver injury and fibrosis in rodents [17,18,25]. Although the mechanical stimulation of CCl_4_ differs from the natural history of NASH, we used CCl_4_ as a fibrosis accelerator in this model. HSCs are a major source of activated myofibroblasts, and their activation is known to drive fibrosis in the liver [26]. HFHCD and CCl_4_ treatment induced the proliferation and activation of HSCs, which were responsible for the rapid progression of fibrosis in our NASH model.

Consistent with our previous study [15], hepatic steatosis was ameliorated in *BAFF^−/−^* mice compared to in WT mice (Figure 1). Moreover, liver inflammation was attenuated in *BAFF^−/−^* mice compared to in WT mice (Figure 2). Itoh et al. [27,28] have reported that CD11c^+^-activated macrophages induce the formation of hepatic CLSs in murine models and patients with NASH, which is consistent with the findings based on our model. BAFF deficiency reduced the number of macrophages, especially CD11c^+^ M1-type macrophages, that produce inflammatory signals such as TNF-α and iNOS, and CLS formation in the livers of HFHCD/CCl_4_-treated mice (Figure 2). It is widely accepted that the M1/M2 balance of macrophages plays an important role in obesity, metabolic syndrome, and fatty liver disease [29]. Macrophages respond to a cocktail of injury signals, including BAFF, and secrete inflammatory substances that may contribute to NASH development and exacerbation.

One of the most relevant findings of our present study is that BAFF-deficient mice are protected from the development of NASH and the progression of fibrosis (Figure 3), which may contribute to the higher liver weight in *BAFF^−/−^* mice compared to that in WT mice (Figure 1b). Furthermore, we confirmed the effect of BAFF on liver fibrosis in the other two models (Figure 4). One of the key features of NASH development and fibrosis progression is the activation of HSCs [26,30,31,32]. In addition to mouse HSC, we observed that the human HSC cell line, LX-2, did not express BAFF-R in our preliminary study. Although the direct effects of BAFF on HSCs in vitro were not observed in our study, BAFF stimulated macrophages to secrete cytokines and other soluble factors that contribute to fibrosis by activating HSC (Figure 6). Similarly, previous reports have demonstrated that macrophages drive HSC activation through the release of soluble factors, such as cytokines, chemokines, and reactive oxygen species, and promote HSC survival [26,31]. Moreover, HSCs and macrophages interact and remodel the extracellular matrix and immune microenvironment among liver cells in the fibrotic liver [32]. Although the influence of other factors has not been disregarded, NO and TNF-α released from macrophages are partly involved in this process. These data indicate that this inflammatory switch is a definitive step in fibrosis progression.

Recent reports have outlined the heterogeneity of hepatic macrophages in NASH [27,28,33,34,35,36]. Satoh et al. [33] have reported that Ceacam1^+^Msr1^+^Ly6C^−^F4/80^−^Mac1^+^ monocytes, which are segregated-nucleus-containing atypical monocytes, are critical for fibrosis. Furthermore, single-cell RNA sequencing has revealed that specific phenotypes of macrophages characterized by high expression of triggering receptor expressed on myeloid cells 2 were expanding in both mouse models and human NASH. These cells, also called NASH-associated macrophages, have been reported to be linked to CLS aggregates and are correlated with disease severity [34]. Initially, we investigated these cells from the liver in our models using flow cytometry; however, the number of these cells did not differ between WT and *BAFF^−/−^* mice. Further studies on the functional properties of these cells are necessary in this regard.

The adipose tissue is also a key factor in the pathogenesis of NAFLD. We previously showed that oxidative stress in visceral adipose tissue, which is one of the underlying causes of NAFLD, regulated serum BAFF levels in HFD-fed mice [14]. Although we did not investigate the VAT in this model, similar mechanisms may play a role in regulating the BAFF level.

We primarily focused on the metabolic aspects of BAFF in the liver; however, BAFF further plays an important role in regulating the immune system [37]. Recently, B lymphocytes have been reported as central mediators of the progression of NAFLD and NASH-associated hepatocellular carcinoma [25,38,39,40,41]. Most of these reports have suggested the pathogenic role of B cells in NAFLD. Interestingly, in mice fed a methionine-choline-deficient diet, B2-cell depletion, including antibody-mediated BAFF neutralization, ameliorated parenchymal damage and lobular inflammation [40]. Furthermore, Novobrantseva et al. [25] have reported that B cells play an important antibody-independent role in tissue repair following liver injury and the development of liver fibrosis. Therefore, it is conceivable that B-cell-directed therapies, including B-cell depletion and approaches targeting B-cell inflammatory mediators such as TNF-α, may ameliorate NASH progression [38]. However, Karl et al. [41] have recently reported that B cells have both detrimental and protective effects in a NAFLD mouse model. Therefore, future studies are warranted to investigate the immunological roles of BAFF and B-cell compartments prior to clinical trials.

In summary, we demonstrated that BAFF depletion ameliorates NASH development and fibrosis progression. Although more preclinical evidence is required, our data suggest that targeting BAFF may be beneficial in treating NASH.

## 4. Materials and Methods

### 4.1. Animals

All study protocols complied with the guidelines of Ehime University (Ehime, Japan), and the protocol was approved by Ehime University Animal Research (No. 05TI70-16).

Male C57BL/6J WT mice and *BAFF^−/−^* mice were purchased from CLEA Japan (Tokyo, Japan) and the Jackson Laboratory (Bar Harbor, ME, USA), respectively. They were maintained at the Department of Biological Resources, Integrated Center for Science, Ehime University, under controlled temperature, humidity, and light (12-h light/dark cycles).

Six-week-old mice, WT mice, and *BAFF^−/−^* mice were fed HFHCD (20 g% fat, 18 g% cholesterol, 22 g% protein, and 45 g% carbohydrates; 1883.7 kJ [450 kcal]/100 g; D16010101; Research Diets, New Brunswick, NJ, USA). They received intraperitoneal injections of CCl_4_ (Wako, Osaka, Japan) at a dose of 0.4 mL/kg diluted 1:25 in olive oil twice a week for 4 weeks from 9 weeks of age. Mice were maintained on each diet and sacrificed at 13 weeks of age.

In certain experiments, mice were fed with HFHCD for 36 weeks or CDAHFD (35.7 g% fat, 23.1 g% protein, and 26.3 g% carbohydrates; 2176.7 kJ [520 kcal]/100 g; A06071302; Research Diets) for 24 weeks.

Serum and liver samples were stored at −80 °C until use. The liver was submerged in RNA later (Life Technologies, Carlsbad, CA, USA) overnight and stored at −20 °C. Serum AST and ALT levels were measured using a Hitachi 7180 autoanalyzer (Hitachi, Ltd., Tokyo, Japan).

### 4.2. Histological and Morphometric Analysis

Liver tissues were fixed in neutral-buffered formalin and embedded in paraffin. Three-micrometer-thick sections were stained with hematoxylin and eosin, SR, and α-SMA (Thermo Fisher Scientific, Waltham, MA, USA). Histological examination was performed in a blinded manner by two experienced liver pathologists with a histological scoring system for NAFLD [42]. The positive areas of each stain(s) were measured digitally using the ImageJ software (National Institutes of Health, Bethesda, MD, USA). To evaluate the degree of fat accumulation, the livers were fixed with osmium tetroxide (OsO_4_), as described previously [15]. SR-positive areas or α-SMA-positive areas were measured using histological light-microscopy images (10×; 7 sections per animal, *n* = 8 animals/group).

Double staining was performed as follows. Liver tissues were fixed with formalin, embedded in paraffin, and thinly sliced. Thereafter, antigen activation was performed using a citrate-buffered solution (pH 6.0), and iNOS was detected using the primary antibodies, rabbit anti-NOS2 (N-20, sc-651; 1:200 dilution) (Santa Cruz Biotechnology, Inc., Dallas, TX, USA). Subsequently, they were labeled with alkaline-phosphatase-conjugated secondary antibodies and detected using the red chromogen, New Fuchsin (DAKO, Glostrup, Denmark). F4/80 was also detected using a rat anti-mouse F4/80 antibody (1:50) (MCA497; BIO-RAD, Hercules, CA, USA), followed by a peroxidase-conjugated secondary antibody. Finally, it was stained dark blue with cobalt-3,3′-diaminobenzidine (Co-DAB). Co-DAB was prepared by adding 2 mL of 1% CoCl_2_ solution to 100 mL of DAB (Nichirei Biosciences, Tokyo, Japan).

### 4.3. Measurement of Hepatic Triglyceride and Cholesterol

Hepatic triglyceride and cholesterol levels were measured at Skylight Biotech (Akita, Japan) using the Folch technique with Cholestest TG and Cholestest CHO kits (Sekisui Medical, Tokyo, Japan), respectively.

### 4.4. Collagen Content in Hepatic Tissue

The total collagen content in the hepatic tissue was measured using a commercially available kit (QuickZyme Total Collagen Assay; QuickZyme Biosciences, Leiden, Netherlands).

### 4.5. Quantitative Real-Time RT-PCR

RNA was extracted using a RNeasy Plus Mini Kit (Qiagen, Hilden, Germany). Reverse transcription was performed using a High-Capacity cDNA Reverse Transcription kit (Applied Biosystems, Foster City, CA, USA), and real-time RT-PCR analysis was performed using SYBR Green I (Roche Diagnostics, Basel, Switzerland) on a LightCycler^®^ 96 (Roche Diagnostics). Primer sequences and annealing temperatures are provided in Table 1. Gene expression data were normalized to the housekeeping gene encoding hypoxanthine phosphoribosyltransferase (HPRT) 1 and expressed as a ratio of the values obtained for WT mice and control RAW264.7, respectively.

The CellAmpTM Direct TB Green RT-qPCR Kit (TAKARA Bio, Shiga, Japan) was used for primary cultured stellate cells according to the manufacturer’s protocol. Gene expression data were normalized to hypoxanthine phosphoribosyltransferase 1 and expressed as a ratio of the values obtained for the CM from PBS-treated-RAW264.7 cells in addition to LPS (Sigma-Aldrich, St. Louis, MO, USA).

### 4.6. Isolation of Primary HSCs

Primary HSCs were isolated from male C57BL/6 mice using the pronase-collagenase digestion method [43] and were cultured in Dulbecco’s Modified Eagle Medium (DMEM) supplemented with 10% fetal bovine serum (FBS; Merck, Darmstadt, Germany). One day after culturing, the cells were treated with 100 ng/mL murine recombinant BAFF (R&D Systems, Minneapolis, MN, USA) for 24 h.

### 4.7. Isolation of Liver Non-Parenchymal Cells (NPCs) and Flow Cytometric Analysis

Liver NPCs and splenocytes were prepared using the procedure described by Chen et al. [8]. Cell suspensions were pre-incubated with anti-CD16/CD32 (clone 93) to block non-specific FcRγ binding and then stained with mouse monoclonal antibodies against the following: CD11c (BioLegend, San Diego, CA, USA), CD11b (BD Biosciences, Franklin Lakes, NJ, USA), CD45 (BioLegend), and F4/80 (BD Biosciences). Flow cytometry was performed using a Gallios flow cytometer (Beckman Coulter, Tokyo, Japan), and data were analyzed using FlowJo 10.7.2. software (Tree Star, Ashland, OR, USA).

### 4.8. Cell Lines

Mouse macrophage-like cell line RAW264.7 was purchased from DS Pharma Biomedical Japan (Osaka, Japan). RAW264.7 cells were cultured in DMEM with high-glucose, L-glutamine, and sodium pyruvate (Thermo Fisher Scientific), supplemented with 10% FBS and 1% penicillin and streptomycin. The cells were treated with 2 μg/mL murine BAFF-R Fc (Enzo Life Sciences, Plymouth Meeting, PA, USA) or PBS. After 2 h, RAW264.7 cells were treated with 100 ng/mL murine recombinant BAFF or PBS for 24 h.

To assess the soluble factors released from macrophages, RAW264.7 cells were treated with 100 ng/mL murine recombinant BAFF and 100 ng/mL LPS for 20 h, and their supernatants were added to primary cultured HSCs on the day of collection.

### 4.9. NF-κB Activity Assay

Nuclear protein extracts were prepared from RAW264.7 cells using the Nuclear Extract kit (Active Motif, Carlsbad, CA, USA) according to the manufacturer’s protocol. As previously described, NF-κB activation was analyzed with the TransAM NF-κB Family kit (Active Motif) [13,14].

### 4.10. Nitrate and TNF-α Determination

The NO concentration in the culture supernatants was determined using the QuantiChrom Nitric Oxide Assay Kit (BioAssay Systems, Hayward, CA, USA) according to the manufacturer’s protocol. TNF-α concentration in the culture supernatants was measured using the enzyme-linked immunosorbent assay (R&D Systems).

### 4.11. Statistical Analysis

Data were analyzed using the JMP version 11.2.0 software (SAS Institute, Cary, NC, USA). Values are shown as the mean ± standard error of the mean or standard deviation. Normally distributed, skewed data and categorical data were analyzed using unpaired *t*-tests, Mann–Whitney *U* tests, and χ^2^ tests, respectively. Differences were considered statistically significant at *p* < 0.05.

## Figures and Tables

**Figure 1 ijms-24-02509-f001:**
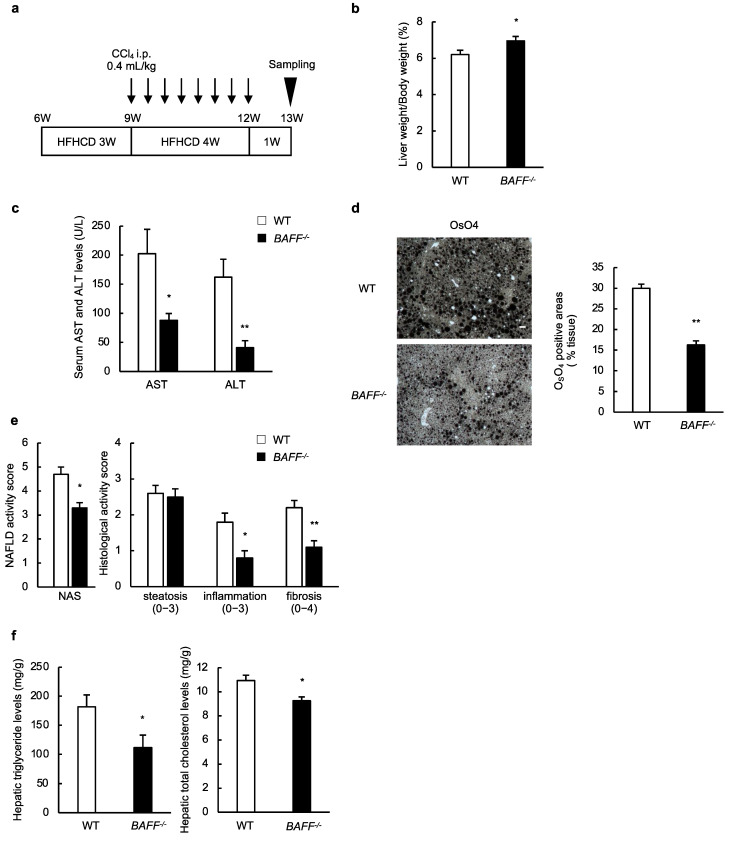
BAFF deficiency attenuates liver steatosis in murine models of non-alcoholic steatohepatitis (NASH). (**a**) Experimental protocol of the new NASH mouse model using HFHCD/CCl_4_-induced liver injury. Six-week-old mice, WT mice and *BAFF^−/−^* mice, were fed with HFHCD. They received intraperitoneal (i.p.) injections of CCl_4_ at a dose of 0.4 mL/kg (diluted 1:25 in olive oil), twice a week for 4 weeks from 9 weeks age; (**b**) Liver weight: (liver weight/body weight) × 100% from WT and *BAFF^−/−^* mice treated with HFHCD/CCl_4_ (*n* = 15/group); (**c**) Serum AST and ALT levels of WT and *BAFF^−/−^* mice treated with HFHCD/CCl_4_ (*n* = 8/group); (**d**) Representative OsO_4_ fixtation (left) and quantification of the OsO_4_-positive area (right) of liver from HFHCD/CCl_4_-treated WT and *BAFF^−/−^* mice (scale bar, 100 μm; images from 8 different fields; *n* = 5/group); (**e**) NAFLD activity score and fibrosis stage of HFHCD/CCl_4_-treated WT and *BAFF^−/−^* mice (*n* = 10/group); (**f**) The levels of hepatic triglyceride (left) and total cholesterol (right) of WT and *BAFF^−/−^* mice treated with HFHCD/CCl_4_ (*n* = 6/group). For all bar plots shown, data represent mean ± standard error of the mean. * *p* < 0.05 and ** *p* < 0.01. BAFF, B cell-activating factor; HFHCD, high-fat/high-cholesterol diet; WT, wild-type; W, weeks; AST, aspartate aminotransferase; ALT, alanine transaminase; OsO_4_, osmium tetroxide; NAS, non-alcoholic fatty liver disease activity score.

**Figure 2 ijms-24-02509-f002:**
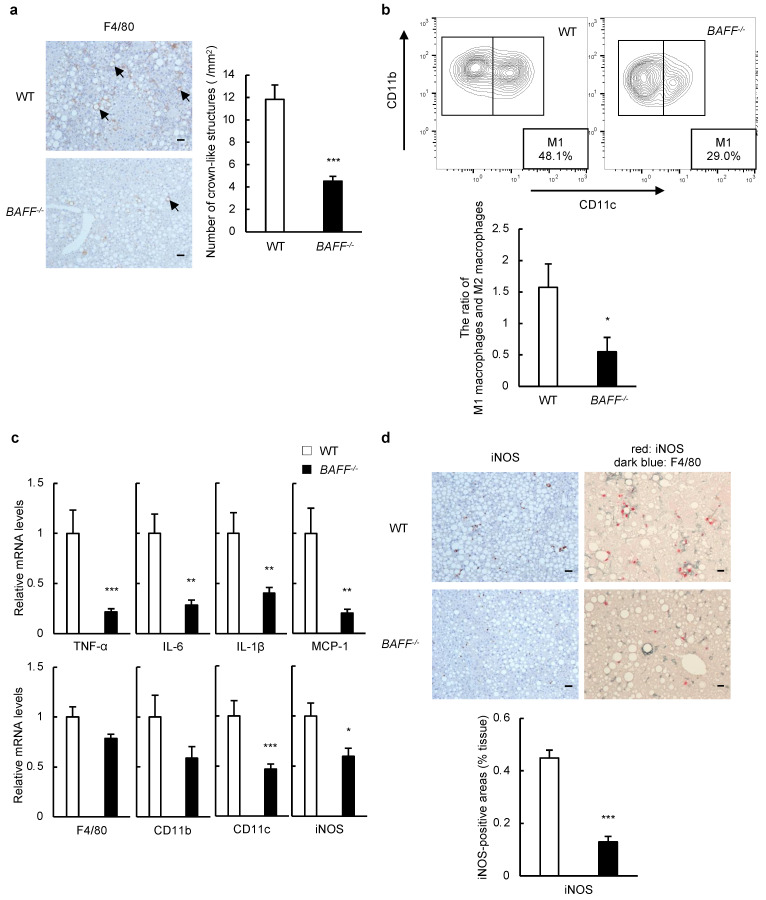
Liver inflammation is attenuated in *BAFF^−/−^* mice in murine models of NASH. (**a**) Representative F4/80 staining of the liver from HFHCD/CCl_4_-treated WT and *BAFF^−/−^* mice (left). The arrows point to the CLSs formed by macrophage aggregation. Quantitative measurement of CLSs (right) (scale bar, 100 μm; images from six different fields; *n* = 8/group); (**b**) Representative flow cytometry contour plots showing frequencies of F4/80^+^CD11b^+^CD11c^+^ cells (M1 macrophages) among liver NPCs isolated from liver from HFHCD/CCl_4_-treated WT and *BAFF^−/−^* mice (upper). The ratio of quantification of M1 macrophages and F4/80^+^CD11b^+^CD11c^−^ cells (M2 macrophages) isolated from liver NPCs (lower) (*n* = 7/group); (**c**) Relative mRNA levels of the indicated genes in the liver from HFHCD/CCl_4_-treated WT and *BAFF^−/−^* mice (*n* = 10–15/group); (**d**) Representative iNOS and F4/80 staining (upper) and quantification of the iNOS-positive area (lower) of liver from with HFHCD/CCl_4_-treated mice. The iNOS-positive macrophages were detected as reddish and dark bluish (scale bar, 100 μm; images from seven different fields; *n* = 5/group). For all bar plots shown, data represent mean ± standard error of the mean. * *p* < 0.05, ** *p* < 0.01, and *** *p* < 0.001. BAFF, B cell-activating factor; NASH, non-alcoholic steatohepatitis; HFHCD, high-fat/high-cholesterol diet; WT, wild-type; CLSs, crown-like structures; NPCs: non-parenchymal cells; TNF, tumor necrosis factor; IL, interleukin; MCP-1, monocyte chemotactic protein-1; iNOS, inducible nitric oxide synthase.

**Figure 3 ijms-24-02509-f003:**
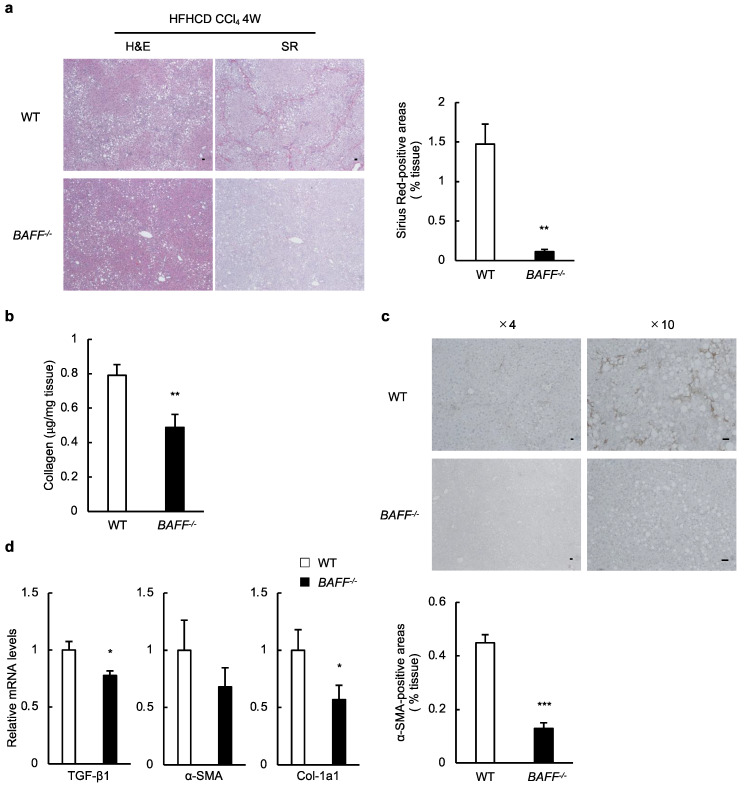
BAFF-deficient mice are protected from the development of NASH and the progression of liver fibrosis. (**a**) Representative H&E and SR staining (left) and quantification of SR-positive area (right) in livers from WT and *BAFF^−/−^* mice treated with HFHCD/CCl_4_ (scale bar, 100 μm; images from seven different fields; *n* = 7/group); (**b**) Total collagen content in the liver was determined using the hydroxyproline assay from HFHCD/CCl_4_-treated mice (*n* = 10/group); (**c**) Representative α-SMA staining (upper) and quantification of α-SMA-positive area of the livers (lower) of HFHCD/CCl_4_-treated mice (scale bar, 100 μm; images from seven different fields; *n* = 8/group); (**d**) Expression of genes related to fibrosis in the livers of HFHCD/CCl_4_-treated mice (*n* = 9/group). For all bar plots shown, data represent mean ± standard error of the mean. * *p* < 0.05, ** *p* < 0.01, and *** *p* < 0.001. BAFF, B cell-activating factor; NASH, non-alcoholic steatohepatitis; H&E, hematoxylin, and eosin; SR, Sirius red; WT, wild-type; HFHCD, high-fat/high-cholesterol diet; W, weeks; α-SMA, α-smooth muscle actin; TGF, transforming growth factor; Col, collagen.

**Figure 4 ijms-24-02509-f004:**
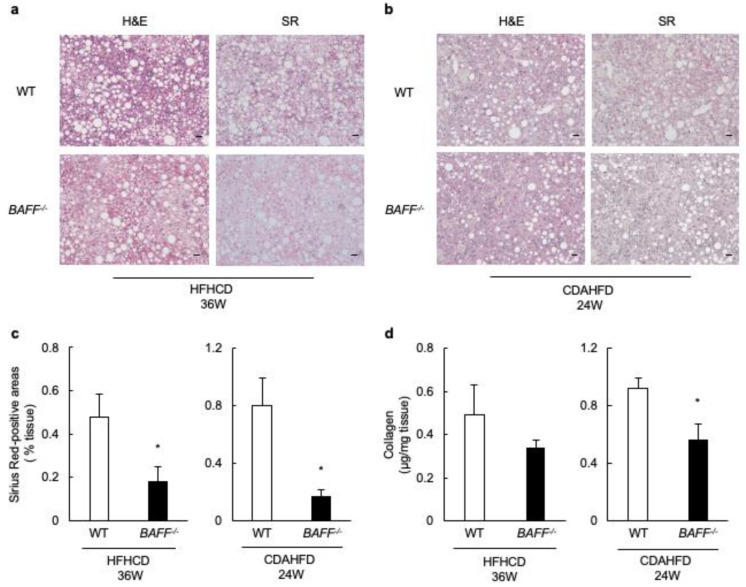
Liver fibrosis is reduced in the absence of BAFF in other models. (**a**,**b**) Representative H&E and SR staining of livers of WT and *BAFF^−/−^* mice fed HFHCD for 36 weeks (**a**) and those fed CDAHFD for 24 weeks (**b**); (**c**) Quantification of the SR-positive areas of livers of WT and *BAFF^−/−^* mice fed each diet (scale bar, 100 μm; images from seven different fields; *n* = 7–13/group); (**d**) Total collagen content in the liver was determined by a hydroxyproline assay from mice fed each diet group (*n* = 5–9/group). For all bar plots shown, data represent mean ± standard error of the mean. * *p* < 0.05. BAFF, B cell-activating factor; H&E, hematoxylin and eosin; SR, Sirius red; HFHCD, high-fat/high-cholesterol diet; CDAHFD, choline-deficient, L-amino-acid-defined high-fat diet; WT, wild-type; W, weeks.

**Figure 5 ijms-24-02509-f005:**
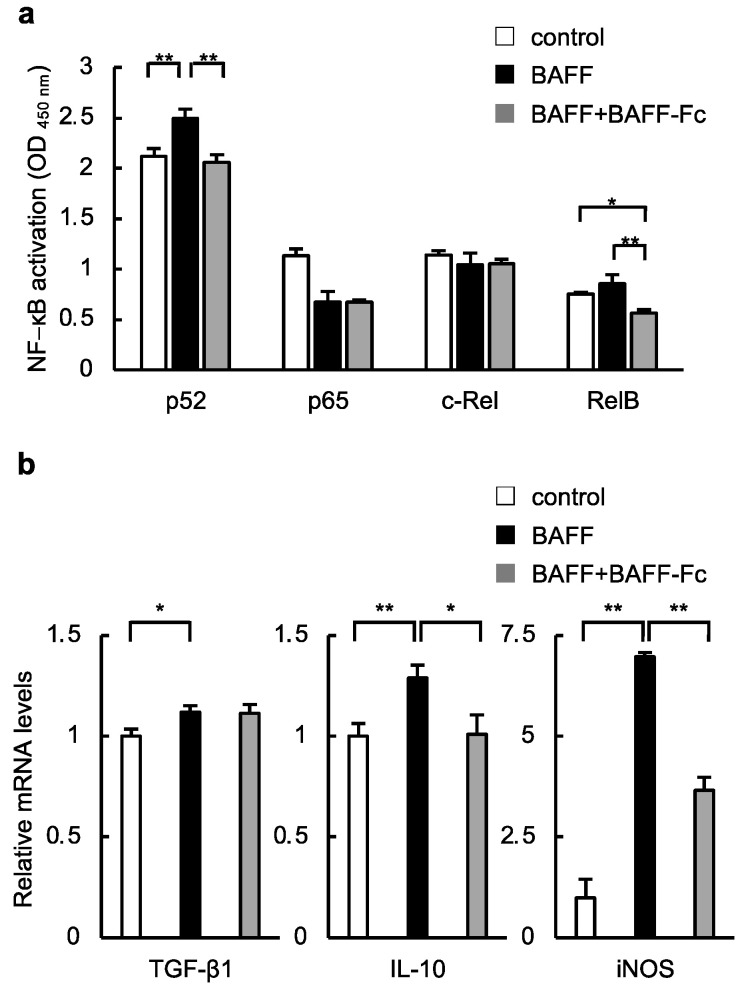
Effect of BAFF on macrophages in vitro. (**a**) The induction of NF-κB in RAW264.7 cells treated with BAFF with or without BAFF-Fc (*n* = 6/group); (**b**) Relative mRNA levels of the indicated genes in RAW264.7 cells treated with BAFF and BAFF-Fc (*n* = 5/group). For all bar plots shown, data represent mean ± standard error of the mean. * *p* < 0.05 and ** *p* < 0.01. BAFF, B cell-activating factor; NF, nuclear factor; TGF, transforming growth factor; IL, interleukin; iNOS, inducible nitric oxide synthase.

**Figure 6 ijms-24-02509-f006:**
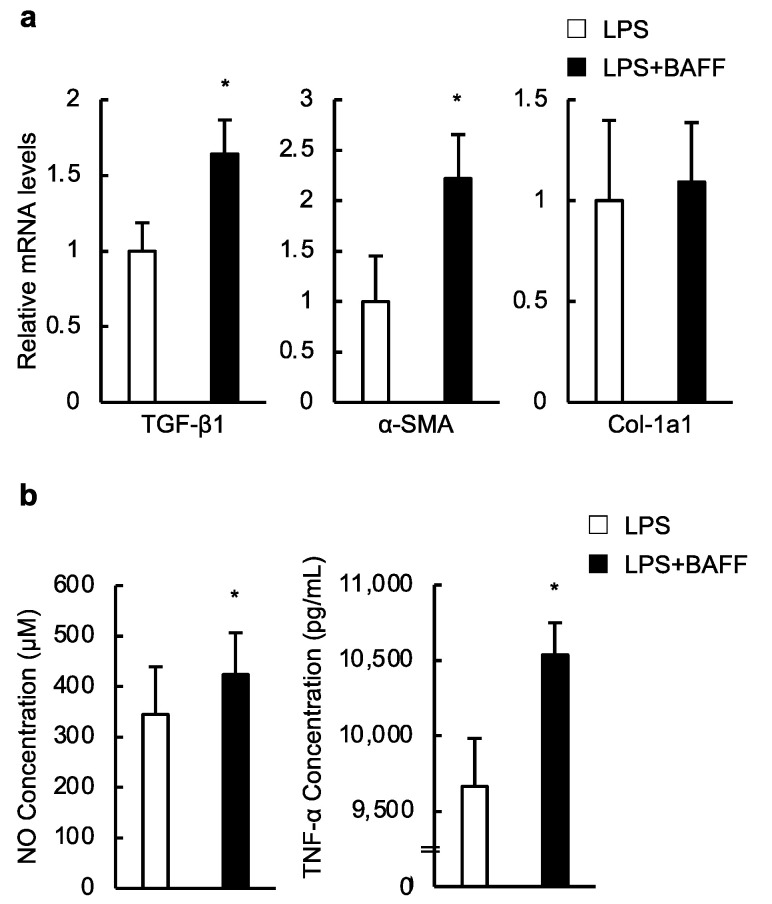
BAFF-treated macrophages generate inflammatory cytokines and activate hepatic stellate cells (HSCs). (**a**) Expression of genes on HSCs following the culture with CM from BAFF and LPS-treated-RAW264.7 cells (*n* = 8–13/group); (**b**) NO (left: *n* = 13/group) and TNF-α concentration (right: *n* = 7/group) in CM from BAFF-treated RAW264.7 cells in addition to LPS were analyzed by parameter NO assay and ELISA, respectively. For all bar plots shown, data represent mean ± standard error of the mean. * *p* < 0.05. BAFF, B cell-activating factor; CM, conditioned medium; LPS, lipopolysaccharide; TGF, transforming growth factor; α-SMA, α-smooth muscle actin; Col, collagen; NO, nitric oxide; TNF, tumor necrosis factor; ELISA: enzyme-linked immunosorbent assay.

**Table 1 ijms-24-02509-t001:** Oligonucleotide sequences and annealing temperature for quantitative real-time PCR.

Primer	Gene Bank Accession no.	Sequence	Location	Annealing
		5′-forward-3′		Temperature
		5′-reverse-3′		(°C)
HPRT1	NM_013556	tcctcctcagaccgcttt	nt104–122	60
		cctggttcatcatcgctaatc	nt173–193	
TNF-α	NM_013693	aggcggtgcctatgtctcag	nt249–268	59
		atgagagggaggccatttggg	nt359–379	
IL-6	NM_031168.2	acaaccacggccttccctactt	nt139–152	63
		cacgatttcccagagaacatgtg	nt245–267	
IL-1β	NM_008361	ttgacggaccccaaaagat	nt142–160	60
		agctggatgctctcatcagg	nt195–214	
MCP-1	NM_011333.3	caggtccctgtcatgcttct	nt89-108	59
		gtggggcgttaactgcat	nt162-179	
F4/80	NM_010130	agtacgatgtggggcttttg	nt35–54	60
		ccccatctgtacatcccact	nt77–96	
CD11b	NM_001082960	cagttcccagaggctctca	nt509–527	65
		cggagccatcaatcaagaag	nt559–578	
CD11c	NM_021334	atggagcctcaagacaggac	nt1725-1744	60
		ggatctgggatgctgaaatc	nt1768-1787	
iNOS	NM_010927	tgaacttgagcgaggagca	nt3518–3540	67
		ttcatgataacgtttctggctct	nt3473–3491	
TGF-β1	NM_011577	tggagcaacatgtggaactc	nt1358–1377	60
		cagcagccggttaccaag	nt1411–1428	
α-SMA	NM_007392	cccacccagagtggagaa	nt9–26	65
		acatagctggagcagcgtct	nt55–74	
Col-1a1	NM_007742	caggcaagcctggtgaac	nt2030–2047	65
		aacctctctcgcctcttgc	nt2089–2107	
IL-10	NM_010548	ggttgccaagccttatcgga	nt302–321	63
		acctgctccactgccttgct	nt473–492	

HPRT1: hypoxanthine phosphoribosyltransferase 1; TNF: tumor necrosis factor; IL: interleukin; MCP-1: monocyte chemotactic protein-1; iNOS: inducible nitric oxide synthase; TGF: transforming growth factor; SMA: smooth muscle actin; Col: collagen.

## Data Availability

The datasets used and/or analyzed during the current study are available from the corresponding author upon reasonable request.

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
