# Peer review of "Role of B Cell-Activating Factor in Fibrosis Progression in a Murine Model of Non-Alcoholic Steatohepatitis"

_ijms, 2023, doi:10.3390/ijms24032509_

Round 1

Reviewer 1 Report

This is a good investigation for improving and expending the potential treatment of liver fibrosis and NAFLD. However, there are some parts of this study can be improved.

First, the histological images of H&E staining in Fig. 1d (left) and Fig. 3a (middle) seems repeatedly used. Please replace the repeatedly used images.

Second, this study using Sirius red staining to present the fibrosis levels. If it's possible, please add trichrome staining in the parts that presenting fibrosis levels.

Third, the nitric oxide levels showed significantly difference between LPS and LPS+BAFF group (Fig. 6b, left). That SEM difference between these two groups seems not able to reach significantly difference. Please provide the original value of this chart. 

Reviewer 2 Report

Kanemitsu-Okada et al. established a modified murine model trying to recapitulate the full spectrum of pathogenesis, including the fibrosis, in NAFLD/NASH. They demonstrated that depletion of BAFF in mice (BAFF-/-) can significantly attenuate high-fat/high-cholestrtrol diet (HFHCD) and carbon tetrachloride (CCl4) induced steatosis, inflammation and fibrosis in liver. Thus, further implying the potential value of using BAFF as a new therapeutic target for treating NAFLD/NASH patients.

Major comments:

1. It's unclear how the current murine model for NASH differs from the existing ones. Specifically, comparison to the similar murine model treated with high-fat diet and CCl4 [18]. It would be helpful to include a table/ schematic showing interventions and phenotypes of all mentioned models in the article.

2. Although normal diet (ND) fed WT and BAFF-/- mice are mentioned in both result 2.1 (line72-74) and method section, however, the corresponding results are missing from the study. They are valued control group and can provide more insights when compared with HFHCD/CCl4 treated mice. 

3. To make it more relevant to human disease, it's necessary to show how   BAFF level is regulated when introducing HFHCD/CCl4 in WT.

4. (Figure 2b) Is the total number of macrophages similar between WT and BAFF-/-?   How about M2 (profibrotic macropaghes) population? 

5. It's a interesting observation that cultured mouse hepatic stellate cells (HSCs) do not express BAFF receptor. Is this also true in vivo or human? It's critical for determining the mechanistic study in fibrosis of human NASH. 

6. (Figure 5) A genetic profiling of BAFF treated macrophages seems more adequate to illustrate the underlining signaling pathway instead of cherry-picking NF-kB as the main focus. 

7. (Figure 6) LPS is considered to induce pro-inflammatory M1, while IL4 and IL10 are often used to induce pro-fibrotic M2. To study HSCs activation in fibrosis, stimulation with condition medium from M2 would be more pathogenic relevant. 

8. As mentioned in #5, if non-direct effects of BAFF on HSCs mediated by macrophages are true in human NASH, then further investigation on responsible mediators from BAFF stimulated macrophages for HSCs activation is critical to demonstrate the role of BAFF in liver fibrosis.

Minor comments:

1. In figure 3a, ND histology was showed without mentioning in the content. 

2. What are the levels in active form of cytokines, such as TGF-b1, IL-10, IL-6, TNF-a, iNOS etc.
